# Systematics, Ecology, and Host Switching: Attributes Affecting Emergence of the Lassa Virus in Rodents across Western Africa

**DOI:** 10.3390/v12030312

**Published:** 2020-03-14

**Authors:** Ayodeji Olayemi, Elisabeth Fichet-Calvet

**Affiliations:** 1Natural History Museum, Obafemi Awolowo University, Ile Ife HO220005, Nigeria; aolayemi@oauife.edu.ng; 2Department of Virology, Bernhard Nocht Institute for Tropical Medicine, 20359 Hamburg, Germany

**Keywords:** Lassa fever, Lassa virus, rodent reservoirs, rodent abundance, phylogeography, virus evolution, endemic hotspots

## Abstract

Ever since it was established that rodents serve as reservoirs of the zoonotic Lassa virus (LASV), scientists have sought to answer the questions: which populations of rodents carry the virus? How do fluctuations in LASV prevalence and rodent abundance influence Lassa fever outbreaks in humans? What does it take for the virus to adopt additional rodent hosts, proliferating what already are devastating cycles of rodent-to-human transmission? In this review, we examine key aspects of research involving the biology of rodents that affect their role as LASV reservoirs, including phylogeography, demography, virus evolution, and host switching. We discuss how this knowledge can help control Lassa fever and suggest further areas for investigation.

## 1. Introduction

Lassa fever, a viral rodent-borne disease, affects thousands of humans in several countries across Western Africa, killing up to 810 in Nigeria alone during 2019 [1]. Consequently, there has been great interest in facets of rodent biology that influence maintenance of the Lassa virus (LASV) in these animals. These topics include which rodents are able to serve as competent LASV hosts at the species level; what combinations of LASV prevalence and rodent abundance across seasons and habitats present the greatest risk to humans; and how LASV mutations are leading to novel virus–rodent relationships that threaten to open up new theatres of rodent-to-human transmission.

Lassa fever was first described as a disease in humans in 1969 within Nigeria [2]. By 1974, the Natal multimammate mouse *Mastomys natalensis* had been identified in Sierra Leone as the natural LASV reservoir [3]. *M. natalensis* is very similar in external morphology to its sibling species and even those from some other genera, thus there has been an increased emphasis on genetic identification of rodent specimens by DNA sequencing and karyotyping in Lassa ecological studies. Subsequent to the initial study in Sierra Leone, LASV surveys incorporating molecular taxonomy of rodents across vast regions including Guinea [4] and southern Mali [5] helped entrench the idea that *M. natalensis* was the sole LASV reservoir. However, contemporary evidence now demonstrates there are alternate LASV rodent hosts. Still, *M. natalensis* is regarded as the major natural LASV host, with most of the published literature on LASV ecology involving this species.

*M. natalensis* is the most widespread rodent species across Sub-Saharan Africa [6,7,8], but only particular populations of this rodent within Western Africa have been found to carry LASV. As the occurrence of Lassa fever in humans is closely linked to the geographic distribution of the virus in *M. natalensis*, there is great curiosity concerning the potential for LASV to spread to currently naïve populations of this rodent within and outside Western Africa.

## 2. LASV Occurrence and Phylogeography of the Rodent *Mastomys natalensis*

Various hypotheses have been proffered concerning what appears to be the proclivity of LASV for certain rodent populations. Indeed, phylogeographic surveys have revealed that each clade of *M. natalensis* tends to host particular arenaviruses. For instance, *M. natalensis* clade A-I, stretching from Guinea up to the Niger river in Western Africa, carries LASV (Figure 1); A-II, distributed from the Niger river into Central Africa, carries a Mobala-like arenavirus; B-IV, in Kenya and Tanzania within East Africa, carries Gairo and Morogoro viruses; while B-VI carries Mopeia and Luna viruses in various countries in southeastern Africa [9,10,11]. Testing the specificity of arenaviruses to *M. natalensis* clades found within Nigeria, LASV was detected in clade A-I to the west of the Niger river and a novel Mobala-like arenavirus in clade A-II to the east [12] (Figure 1). However, although clade A-II did not carry LASV in eastern Nigeria, A-II specimens found in the hybrid zone along the western bank of the Niger river were infected with LASV. These results demonstrate that although A-II was not found to carry LASV in eastern Nigeria, it is not resistant to infection and the virus could well emerge eventually in this *M. natalensis* clade, which extends east of Nigeria into Cameroon and other countries in Central Africa.

LASV occurrence is not only disparate between different *M. natalensis* clades, but also within. Two major hotspots of LASV infection occur across Western Africa, one in Nigeria and the other in the Mano River Union (MRU) countries (Guinea, Sierra Leone, and Liberia) (Figure 1). Few other foci appear in places such in southern Mali and northern Ivory Coast [5,13,14] (Figure 1). Attempting to explain this fragmentation, risk map models indicate some correlation between LASV occurrence and environmental variables such as rainfall and temperature [11,15,16,17]. Recently, social components such as human connectivity have been added in the spatial model, which predicted the recent outbreak in Nigeria [18]. However, even within well-established LASV hotspots, the local distribution is spotted, with probable cycles of viral colonization and extinction driven by human-aided movement or fluctuation in rodent populations. 

## 3. Fluctuation in LASV Prevalence and *M. natalensis* Abundance

In Western Africa, there are two annual seasons, namely wet (from April to October) and dry (from November to March). Of these, Lassa fever outbreaks in humans occur more significantly during the height of the dry season [1,19]. There is curiosity if this periodicity is linked in any way to virus–rodent dynamics. In Guinea, a significant increase in LASV prevalence was recorded in *M. natalensis* during the rainy season compared to the dry season, whereas a statistically higher abundance of this rodent species was found inside human habitations compared to other outdoor sites during the dry season [20]. This suggests that rodent abundance indoors (even with a lower LASV prevalence) is probably a stronger factor contributing to dry season Lassa fever outbreaks.

In a similar study within Nigeria, LASV prevalence did not fluctuate significantly between rainy and dry seasons in the rodent *M. natalensis* in the highly endemic Edo area, Nigeria (likely due to inadequate sample size) [21]. Nevertheless, specimens collectively across various localities had a statistically higher distribution indoors over outdoor habitats throughout the year [22]. This shows that the commensality of *M. natalensis* plays a central role in LASV rodent-to-human transmission. Hence, how possible is it to significantly reduce levels of rodent infestation inside and around human houses? Recently, the authors of a four-year study in Guinea observed that for rodent control using chemical bait to be effective in the long term, it has to be combined with general environmental hygiene, coordinated across whole villages instead of select addresses, continuous through time (as *Mastomys* populations can recover within a few months), and enhanced by a higher quality of domestic amenities (e.g., rodent-proof walls, roofs, and food storage) [23]. 

## 4. The Ability of LASV to Switch between Different *Mastomys* Rodent Species

Forty-seven years after Lassa fever was first described as a disease, the isolation of LASV has now been reported from rodent hosts other than *Mastomys natalensis*. These new LASV reservoirs are the Guinea multimammate mouse *Mastomys erythroleucus* in both Nigeria and Guinea [24], the African wood mouse *Hylomyscus pamfi* in Nigeria [24], and the Pygmy mouse *Mus baoulei* in Ghana and Benin [25,26]. Among them, the more epidemiologically important appears to be *M. erythroleucus*, in terms of the geographic range and diversity of LASV lineages hosted by populations of this species. LASV lineage IV was found in *M. erythroleucus* in Madina Oula (Figure 1), coastal Guinea, where a heavy LASV human seroprevalence was recorded in the 1990s [27]. Among the 16 animals caught in that locality, 6 were LASV positive and 7 were IgG positive (IFA test), leading to an infection rate of 75%. It is now apparent that *M. erythroleucus* rather than *M. natalensis* is the key LASV reservoir in this part of Guinea.

The cerebrospinal fluid (CSF) strain of LASV lineage III (so named after it was detected in the cerebrospinal fluid of a human patient [28]) was found in *M. erythroleucus* to the immediate south of the Benue river, but still in the vicinity where lineage III is known to circulate in central Nigeria [29]. Following up on this result, preliminary evidence has been provided by Adesina et al. of LASV lineage II detected for the first time in *M. erythroleucus* together with *M. natalensis* in the same locality within the Edo state, southwestern Nigeria [30]. The Edo hotspot (indicated by lineage II within Nigeria in Figure 1) records among the highest number of Lassa fever cases in the country [19] and is where this same LASV lineage has been detected in both humans [19] and *M. natalensis* rodents [12]. The LASV strains Adesina et al. recently found at the same site in both *M. natalensis* and *M. erythroleucus* within Edo were not genetically distinct and overlapped phylogenetically, suggesting horizontal transmission. This demonstrates it is probably easier than previously thought for LASV to jump between these two rodent species. 

Serological evidence additionally shows that the role of *M. erythroleucus* as a LASV host extends down to various localities in southern Nigeria [21]. The Guinea multimammate mouse is normally a savanna species distributed in the northern and central part of the country (Figure 2), but is being increasingly detected in degraded forests and clearings within southern Nigeria [31], illustrating the link between biodiversity loss and zoonotic disease emergence [32].

The increasing epidemiological significance of *M. erythroleucus*, in combination with the already well-known role of *M. natalensis*, may be contributing to the heightened burden of Lassa fever that has been reported in recent years within Nigeria [39]. However, the potential remains for this interaction to gain even further momentum. *M. natalensis* and *M. erythroleucus* are distinct but sister taxa and likely offer “similar biological host environments” for adaptation and multiplication of the virus after a spillover between infected and recipient species [40]. These two rodent species are also ecologically similar, able to live commensally with humans and produce demographic explosions in cultivations in Senegal [41,42] or in Tanzania [7,43]. With a multitude of overlapping populations across Western Africa (Figure 2), mapping out localities where these *Mastomys* species occur sympatrically will help anticipate the LASV emergence that threatens from their concerted zoonotic effect.

## 5. Evolution of Novel LASV Lineages in Non-*Mastomys* Rodents

Expanded surveys at the turn of the 21st century have led to the discovery of a number of novel arenaviruses in rodent populations across Western Africa, including the Kodoko virus in *Mus minutoides* in Guinea [44], the Menekre virus in *Hylomyscus* sp. in Ivory Coast [45], the Natorduori virus in *Mus mattheyi* in Ghana [25], and a Mobala-like arenavirus in *Mastomys natalensis* in Nigeria [12] (Figure 1). Quite striking is a certain category among these newly discovered arenaviruses that are phylogenetically close to LASV lineage I, which is basal to the other lineages (II–V). These LASV-like arenaviruses are the Gbagroube virus found in *Mus setulosus* in southern Ivory Coast [45], the Jirandogo virus in *Mus baoulei* in Ghana and Benin [25,26], and the Kako virus in *Hylomyscus pamfi* in southwestern Nigeria [24]. A feature common to these arenaviruses is that they were detected in non-*Mastomys* within what have up till now been regarded as non-endemic areas for Lassa fever between Nigeria and the MRU countries.

Kako and Jirandogo fall within the phylogenetic scope that defines LASV, clustering to the Lily Pinneo strain (lineage I) and also the new lineages recently derived from humans in Togo and Benin [46,47]. However, the phylogenetic nodes are deeper between these novel lineages, indicating older host switching among the non-*Mastomys* rodents than between the *Mastomys* spp. (Figure 3). The emerging hotspot for Lassa fever in the Togo and Benin area therefore features a distinct ecology, maintained by rodent species other than those belonging to the genus *Mastomys*.

## 6. Conclusions

Four rodent species have now been identified as LASV reservoirs: two commensals, *Mastomys natalensis* and *Mastomys erythroleucus*, and two wild ones, the African wood mouse *Hylomyscus pamfi* and the Pygmy mouse *Mus baoulei* (Figure 2, Appendix A). In Western Africa, the LASV genome is highly variable from one region to another, and there are now 7 lineages: 4 in Nigeria, 1 in Togo, 1 in Guinea-Sierra Leone-Liberia, and 1 in Mali-Ivory Coast [4,5,12,13,24,25,26,30]. Strains that are described in the four species of rodents belong to several lineages: lineages II and IV in *M. natalensis*, lineages II, III, and IV in *M. erythroleucus,* a new lineage in *H. pamfi*, and another new lineage in *M. baoulei* (Figure 3). According to the transmission pattern suggested by Karesh et al. [55], some spillover events could occur between (**a**) wild and commensal reservoirs, (**b**) between wild reservoirs and humans, and (**c**) between commensal reservoirs and humans. The ease with which LASV seems to jump from one species to another as well as the presence of IgG antibodies in other species belonging to the genera *Praomys* or *Lemniscomys* point in the direction of future new reservoir candidates [21,56]. In order to anticipate the full-scale emergence of these new lineages in humans, more intensive small mammal screening, including careful consideration of non-*Mastomys* species, should be sustained across Western Africa.

## Figures and Tables

**Figure 1 viruses-12-00312-f001:**
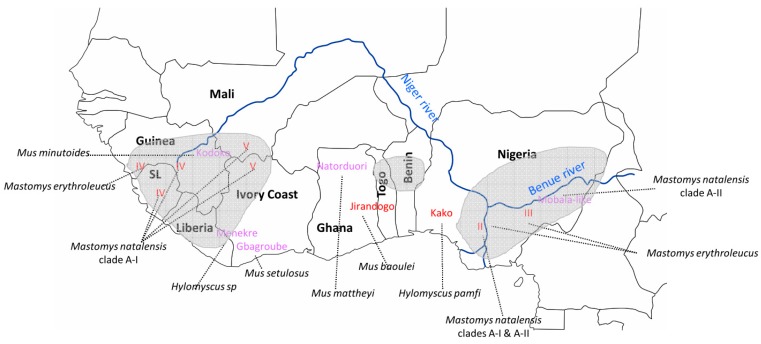
Lassa virus (LASV) lineages (in red) and arenaviruses (in purple) detected in rodents across Western Africa (with host species indicated in italics and connected by dashed lines to corresponding viruses). Endemic foci for Lassa fever are shaded in grey. Respective country names appear within the map in bold black.

**Figure 2 viruses-12-00312-f002:**
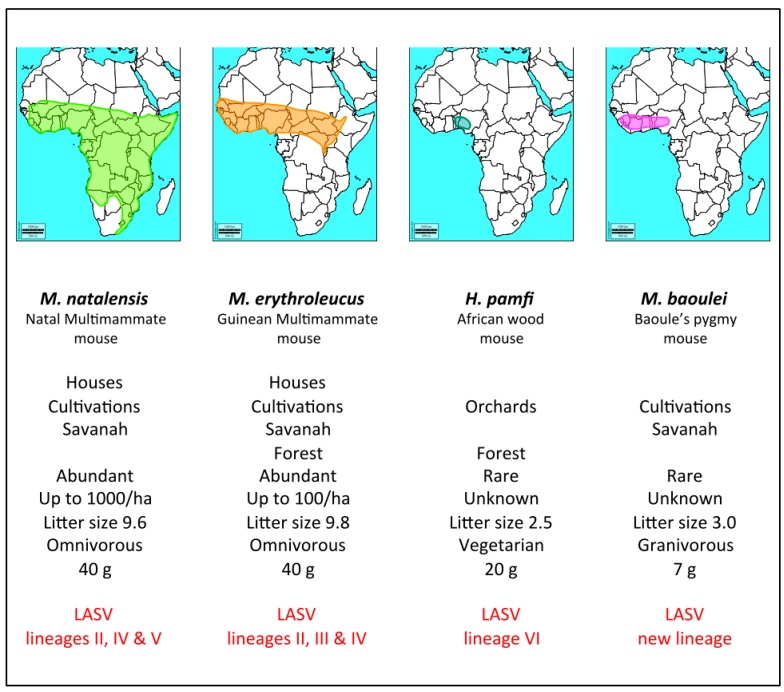
Distribution of the 4 host species: *Mastomys natalensis*, *Mastomys erythroleucus*, *Hylomyscus pamfi*, and *Mus (Nannomys) baoulei* in Africa with their main traits of life. The information was compiled from [4,5,12,13,24,25,26,30,33,34,35,36,37,38] and personal data.

**Figure 3 viruses-12-00312-f003:**
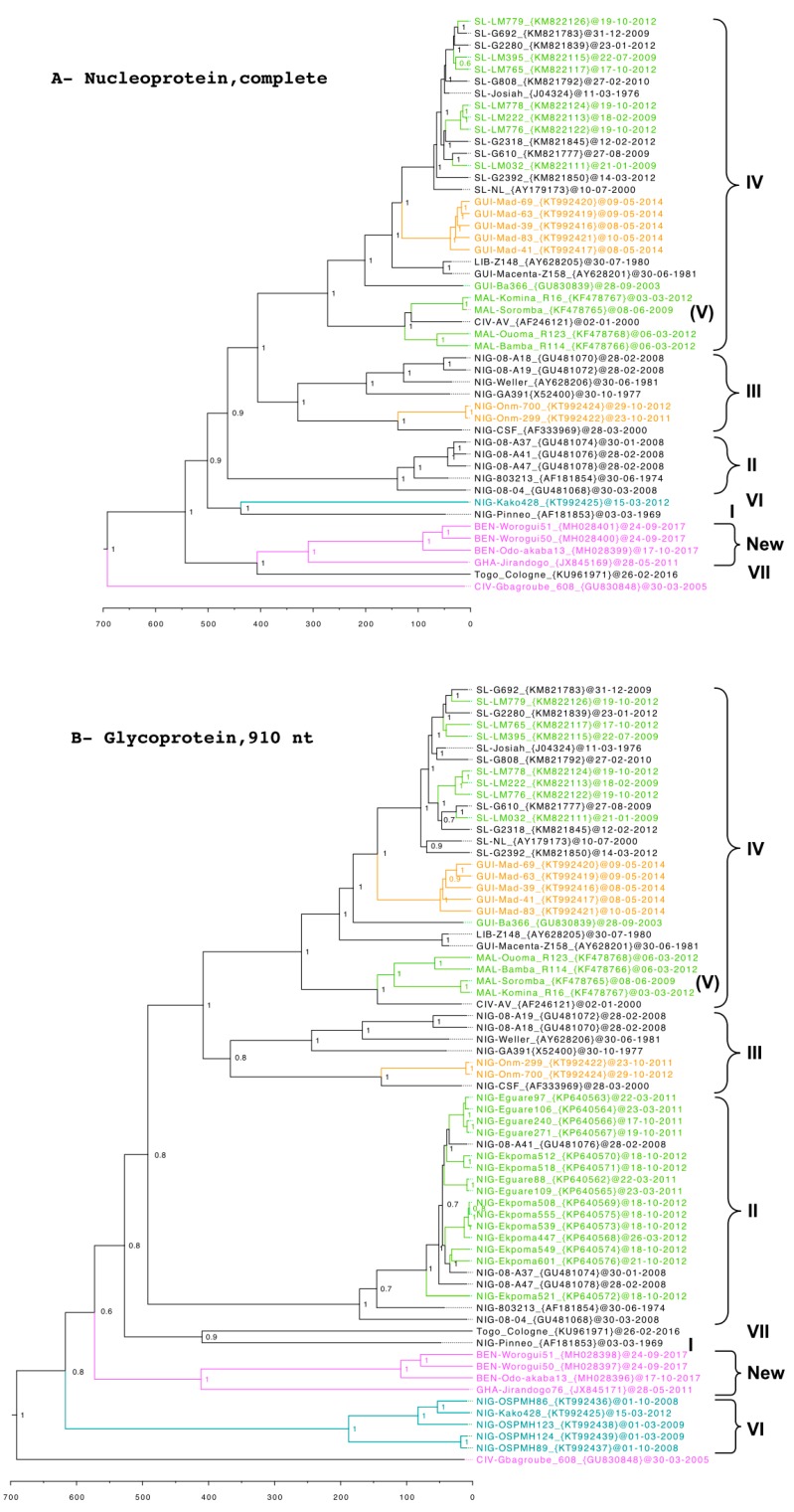
Time-calibrated phylogenetic analysis of LASV in humans (in black), *Mastomys natalensis* (in green), *Mastomys erythroleucus* (in orange), *Hylomyscus pamfi* (in blue), and *Mus baoulei* (in pink). The tree is rooted with the Gbagroube virus found in *Mus setulosus* (in pink). Statistical support of grouping from Bayesian posterior probabilities is indicated at the node. Country, strain name, GenBank accession numbers, and day of collection are indicated on the labels. Scale bar indicates time in years. The analysis was inferred by using the Bayesian Markov Chain Monte Carlo (MCMC) method implemented in BEAST software [48]. The following settings were used: GTR+gamma with codon partition 1,2,3, strict clock and constant population. MCMC chains were run for 10 million states and sampled every 10,000 states to obtain an effective sample size above 200 for all the parameters. (**A**) The tree is based on the complete nucleoprotein NP gene (1710 nucleotides), including 48 taxa. (**B**) The tree is based on the partial glycoprotein GP gene (910 nucleotides), including 67 taxa, in particular those from *M. natalensis* in the Edo state and those from *H. pamfi* in the Osun state. Sequences used for these analyses were published in [5,12,24,28,29,45,46,49,50,51,52,53,54] as well as in Jahrling et al., Direct Submission, GenBank.

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
