# Peer review of "Systematics, Ecology, and Host Switching: Attributes Affecting Emergence of the Lassa Virus in Rodents across Western Africa"

_viruses, 2020, doi:10.3390/v12030312_

Round 1
Reviewer 1 Report
This well-written and timely manuscript describes the systematics, ecology and host-switching propensity of Lassa and LASV-like viruses in the context of their rodent hosts in West Africa, offering a comprehensive overview of what is known regarding ecological factors that contribute to spread and evolution of the virus. The figures are nicely prepared and contribute to our understanding of the text. The work will make a valuable addition to the planned volume.
Author Response
We thank you for your positive appreciation.
Reviewer 2 Report
This review by Olayemi and Fichet-Calvet presents an updated view of the current knowledge about the distribution of Lassa virus (LASV) in different rodent species within Western Africa.
This is a timely review as the implementation of novel sequence technologies has resulted in the identification of a large number of LASV, and other mammarenaviruses, genome sequences in different rodent species in Western Africa. This, in turn, has shown that, contrary to the initial view of the Natal multimammate mouse M. natalensis as the sole rodent species acting as LASV reservoir, other rodent species can also carry LASV. The review has captured the complexity of the ecology of LASV, including its occurrence within different clades of the M. natalensis, and importantly the variable presence of LASV within the same clade of M. natalensis including within known geography LASV hotspots. The comment of LASV’s ability to “jump” between M. natalensis and M. erythroleucus needs some further elaboration, as in its present form the statement appears to suggest a horizontal transmission of LASV between these two rodent species, which has not been proven and it is quite possible that the presence of LASV in M. erythroleucusdoes not reflect horizontal transmission events.
The reading of the paper in its present form might pose some challenges for the non-experts. Incorporation of Table presenting the different rodent species, including clades in the case of M. natalensis, and the associated LASV lineages and other detected mammarenaviruses, combined with an extended version of figure 1 would help the non-experts to capture the complexity of the ecology of LASV and its different rodent reservoirs.
Minor comments:
1) The opening sentence in the introduction states the estimated number of LASV infections in humans that occur yearly in Western Africa based on information provided by a rather old citation that needs to be reconsidered, as it represents extrapolations from a single longitudinal study conducted more than 30 years ago in Sierra Leone, whereas the true incidence of LASV infections and public health burden of LF in Western Africa are unknown.
2) It would be better using the standardized UN Geoscheme nomenclature for geographic designations (e.g. Western Africa rather than West Africa)
3) Figure 2. Adding the pictures of the rodents may be a good option to capture the interest.
4) Page 3, line 87: Edo area needs additional description. where is Edo area? Some explanation (ex. country or regions) is helpful.
5) Page 5, lines 151-153: this sentence needs additional explanation.
6) Figure 3: the selected color code makes it very difficult to appreciate some of the names.
7) Page 7, Line 175: add references after the “Mali-Ivory Coast”.
Author Response
This review by Olayemi and Fichet-Calvet presents an updated view of the current knowledge about the distribution of Lassa virus (LASV) in different rodent species within Western Africa.
This is a timely review as the implementation of novel sequence technologies has resulted in the identification of a large number of LASV, and other mammarenaviruses, genome sequences in different rodent species in Western Africa. This, in turn, has shown that, contrary to the initial view of the Natal multimammate mouse M. natalensis as the sole rodent species acting as LASV reservoir, other rodent species can also carry LASV. The review has captured the complexity of the ecology of LASV, including its occurrence within different clades of the M. natalensis, and importantly the variable presence of LASV within the same clade of M. natalensis including within known geography LASV hotspots. The comment of LASV’s ability to “jump” between M. natalensis and M. erythroleucus needs some further elaboration, as in its present form the statement appears to suggest a horizontal transmission of LASV between these two rodent species, which has not been proven and it is quite possible that the presence of LASV in M. erythroleucus does not reflect horizontal transmission events.
This section has been elaborated further as follows: “Following up on this result, preliminary evidence has been provided by Adesina et al. of LASV lineage II detected for the first time in M. erythroleucus together with M. natalensis in the same locality within Edo state, southwestern Nigeria [30]. The Edo hotspot (indicated by lineage II within Nigeria in Figure 1) records among the highest number of Lassa fever cases in the country [19] and is where this same LASV lineage has been detected in both humans [19] and only M. natalensis rodents [12]. The LASV strains Adesina et al. recently found in the same site in both M. natalensis and M. erythroleucus within Edo were not genetically distinct and overlapped phylogenetically, suggesting horizontal transmission. This demonstrates it is probably easier than previously thought for LASV to jump between these two rodent species.”
We think the phylogenetic analysis carried out by our team in Adesina et al. [30], which shows no genetic distinction between LASV strains derived from M. natalensis and M. erythroleucus rodents, supports the idea that horizontal transmission takes place between individuals of both species.
The reading of the paper in its present form might pose some challenges for the non-experts. Incorporation of Table presenting the different rodent species, including clades in the case of M. natalensis, and the associated LASV lineages and other detected mammarenaviruses, combined with an extended version of figure 1 would help the non-experts to capture the complexity of the ecology of LASV and its different rodent reservoirs.
Figure 2 represents an illustration that summarizes the complexity of the ecology of LASV and its different rodent reservoirs. We have now expanded this figure to include information concerning which LASV lineages are hosted by which rodents.
Minor comments:
1) The opening sentence in the introduction states the estimated number of LASV infections in humans that occur yearly in Western Africa based on information provided by a rather old citation that needs to be reconsidered, as it represents extrapolations from a single longitudinal study conducted more than 30 years ago in Sierra Leone, whereas the true incidence of LASV infections and public health burden of LF in Western Africa are unknown.
We now cite more recent data provided by the Nigerian Centre for Disease Control.
2) It would be better using the standardized UN Geoscheme nomenclature for geographic designations (e.g. Western Africa rather than West Africa)
We have now changed “West” to “Western” Africa throughout the text.
3) Figure 2. Adding the pictures of the rodents may be a good option to capture the interest.
We provided a supplementary information gathering photos of rodents during the necropsies performed in Guinea, Ghana and Nigeria. All the animals were identified by cytochrome b sequencing. As the rodents were dead, and therefore not nicely visible, we think that the pictures are more suitable in a supplementary information than in the core of the text.
4) Page 3, line 87: Edo area needs additional description. Where is Edo area? Some explanation (ex. country or regions) is helpful.
“Nigeria” is now added in this line. In addition, we have re-phrased a part of section 4 of the manuscript, giving details on Edo area. See our response concerning the horizontal transmission in the major comments.
5) Page 5, lines 151-153: this sentence needs additional explanation.
The sentence has been re-adjusted for clarity: “However, the phylogenetic nodes are deeper between these novel lineages, indicating older host switching among the non-Mastomys rodents than between the Mastomys spp [Figure 3].”
6) Figure 3: the selected color code makes it very difficult to appreciate some of the names.
The colours are linked with those in the maps on figure 2, and it is difficult to change the codes. Nevertheless, we have accentuated the tone of the light green (probably the less visible) and changed the dark green to blue.
7) Page 7, Line 175: add references after the “Mali-Ivory Coast” [4,5,12,13, 24-26,30].
Effected as above.